# LeRaC: Learning Rate Curriculum

## Abstract

Most curriculum learning methods require an approach to sort the data samples by difficulty, which is often cumbersome to perform. In this work, we propose a novel curriculum learning approach termed **Le**arning **Ra**te Curriculum (LeRaC), which leverages the use of a different learning rate for each layer of a neural network to create a data-free curriculum during the initial training epochs. More specifically, LeRaC assigns higher learning rates to neural layers closer to the input, gradually decreasing the learning rates as the layers are placed farther away from the input. The learning rates increase at various paces during the first training iterations, until they all reach the same value. From this point on, the neural model is trained as usual. This creates a model-level curriculum learning strategy that does not require sorting the examples by difficulty and is compatible with any neural network, generating higher performance levels regardless of the architecture. We conduct comprehensive experiments on eight datasets from the computer vision (CIFAR-10, CIFAR-100, Tiny ImageNet), language (BoolQ, QNLI, RTE) and audio (ESC-50, CREMA-D) domains, considering various convolutional (ResNet-18, Wide-ResNet-50, DenseNet-121), recurrent (LSTM) and transformer (CvT, BERT, SepTr) architectures, comparing our approach with the conventional training regime. Moreover, we also compare with Curriculum by Smoothing (CBS), a state-of-the-art data-free curriculum learning approach. Unlike CBS, our performance improvements over the standard training regime are consistent across all datasets and models. Furthermore, we significantly surpass CBS in terms of training time (there is no additional cost over the standard training regime for LeRaC). Our code is freely available at: `http//github.com/link.hidden.for.review`.

## 1 Introduction

Machine learning researchers relentlessly strive to improve the performance of AI models. Much of this effort has been directed to the development of novel neural architectures [1–9], which have grown in size and complexity [1, 7, 10] to leverage the availability of increasingly larger datasets. However, we believe the dominant trend to develop deeper and deeper neural networks is not sustainable on the long term. To this end, we turn our attention to an alternative approach to increase performance of deep neural models without growing the size of the respective models. More specifically, we focus on curriculum learning, an approach initially proposed by Bengio et al. [11] to train better neural networks by mimicking how humans learn, from easy to hard. As originally introduced by Bengio et al. [11], curriculum learning is a training procedure that first organizes the examples in their increasing order of difficulty, then starts the training of the neural network on the easiest examples, gradually adding increasingly more difficult examples along the way, until all training examples are fed to the network. The success of the approach relies in avoiding to force the learning of very difficult examples right from the beginning, instead guiding the model on the right path through the imposed curriculum. This type of curriculum is later referred to as data-level curriculum learning [12]. Indeed, Soviany et al. [12] identified several types of curriculum learning approaches in the

literature, dividing them into four categories based on the components involved in the definition of machine learning given by Mitchell [13]. The four categories are: data-level curriculum (examples are presented from easy to hard), model-level curriculum (the modeling capacity of the network is gradually increased), task-level curriculum (the complexity of the learning task is increased during training), objective-level curriculum (the model optimizes towards an increasingly more complex objective). While data-level curriculum is the most natural and direct way to employ curriculum learning, its main disadvantage is that it requires a way to determine the difficulty of the data samples. The task of estimating the difficulty of the data samples has been addressed in different domain-specific ways, e.g. the length of text has been used in natural language processing [14, 15], while the number or size of objects were shown to work well in computer vision [16, 17]. Despite having many successful applications [12, 18], there is no universal way to determine the difficulty of the data samples, making the data-level curriculum less applicable to scenarios where the difficulty is hard to estimate, e.g. classification of radar signals. The task-level and objective-level curriculum learning strategies suffer from similar issues, e.g. it is hard to create a curriculum when the model has to learn an easy task (binary classification) or the objective function is already convex.

Considering the above observations, we recognize the potential of model-level curriculum learning strategies of being applicable across a wider range of domains and tasks. To date, there are only a few works [19–21] in the category of pure model-level curriculum learning methods. Furthermore, the existing methods have some drawbacks caused by their domain-dependent or architecture-specific design. For instance, Karras et al. [20] gradually increase the resolution of input images as new layers are appended to a generative network, but the notion of input resolution does not exist in other domains, e.g. text. Burduja et al. [19] blur the input images with Gaussian kernels, but this method is not applicable to an input format for which there is no convolution operation, e.g. tabular data. Sinha et al. [21] apply Gaussian kernel smoothing on convolutional activation maps, but this operation makes less sense for a feed-forward neural network formed only of dense layers.

To benefit from the full potential of the model-level curriculum learning category, we propose LeRaC (**Le**arning **Ra**te **C**urriculum), a novel and simple curriculum learning approach which leverages the use of a different learning rate for each layer of a neural network to create a data-free curriculum during the initial training epochs. More specifically, LeRaC assigns higher learning rates to neural layers closer to the input, gradually decreasing the learning rates as the layers are placed farther away from the input. This prevents the propagation of noise caused by the random initialization of the network's weights. The learning rates increase at various paces during the first training iterations, until they all reach the same value. From this point on, the neural model is trained as usual. This creates a model-level curriculum learning strategy that is applicable to any domain and compatible with any neural network, generating higher performance levels regardless of the architecture, without adding any extra training time. To the best of our knowledge, we are the first to employ a different learning rate per layer to achieve the same effect as conventional (data-level) curriculum learning.

We conduct comprehensive experiments on eight datasets from the computer vision (CIFAR-10 [22], CIFAR-100 [22], Tiny ImageNet [23]), language (BoolQ [24], QNLI [25], RTE [25]) and audio (ESC-50 [26], CREMA-D [27]) domains, considering various convolutional (ResNet-18 [4], Wide-ResNet-50 [28], DenseNet-121 [29]), recurrent (LSTM [30]) and transformer (CvT [8], BERT [2], SepTr [31]) architectures, comparing our approach with the conventional training regime and Curriculum by Smoothing (CBS) [21], our closest competitor. Unlike CBS, our performance improvements over the standard training regime are consistent across all datasets and models. Furthermore, we significantly surpass CBS in terms of training time, since there is no additional cost over the conventional training regime for LeRaC, whereas CBS adds Gaussian kernel smoothing layers.

In summary, our contributions are twofold:

- We propose a novel and simple model-level curriculum learning strategy that creates a curriculum by updating the weights of each neural layer with a different learning rate, considering higher learning rates for the low-level feature layers and lower learning rates for the high-level feature layers.

- We empirically demonstrate the applicability to multiple domains (image, audio and text), the compatibility to several neural network architectures (convolutional neural networks, recurrent neural networks and transformers), and the time efficiency (no extra training time added) of LeRaC through a comprehensive set of experiments.

## 2 Related Work

Curriculum learning was initially introduced by Bengio et al. [11] as a training strategy that helps machine learning models to generalize better when the training examples are presented in the ascending order of their difficulty. Extensive surveys on curriculum learning methods, including the most recent advancements on the topic, were conducted by Soviany et al. [12] and Wang et al. [18]. In the former survey, Soviany et al. [12] emphasized that curriculum learning is not only applied at the data level, but also with respect to the other components involved in a machine learning approach, namely at the model level, the task level and the objective (performance measure) level. Regardless of the component on which curriculum learning is applied, the technique has demonstrated its effectiveness on a broad range of machine learning tasks, from computer vision [11, 16, 17, 21, 32–34] to natural language processing [11, 35–38] and audio processing [39, 40].

The main challenge for the methods that build the curriculum at the data level is measuring the difficulty of the data samples, which is required to order the samples from easy to hard. Most studies have addressed the problem with human input [41–43] or metrics based on domain-specific heuristics. For instance, the length of the sentence [36, 44] and the word frequency [11, 38] have been employed in natural language processing. In computer vision, the samples containing fewer and larger objects have been considered to be easier in some works [16, 17]. Other solutions employed difficulty estimators [45] or even the confidence level of the predictions made by the neural network [46, 47] to approximate the complexity of the data samples.

The solutions listed above have shown their utility in specific application domains. Nonetheless, measuring the difficulty remains problematic when implementing standard (data-level) curriculum learning strategies, at least in some application domains. Therefore, several alternatives have emerged over time, handling the drawback and improving the conventional curriculum learning approach. In [48], the authors introduced self-paced learning to evaluate the learning progress when selecting the easy samples. The method was successfully employed in multiple settings [48–54]. Furthermore, some studies combined self-paced learning with the traditional pre-computed difficulty metrics [53, 55]. An additional advancement related to self-paced learning is the approach called self-paced learning with diversity [56]. The authors demonstrated that enforcing a certain level of variety among the selected examples can improve the final performance. Another set of methods that bypass the need for predefined difficulty metrics is known as teacher-student curriculum learning [57, 58]. In this setting, a teacher network learns a curriculum to supervise a student neural network.

Closer to our work, a few methods [19–21] proposed to apply curriculum learning at the model level, by gradually increasing the learning capacity (complexity) of the neural architecture. Such curriculum learning strategies do not need to know the difficulty of the data samples, thus having a great potential to be useful in a broad range of tasks. For example, Karras et al. [20] proposed to gradually add layers to generative adversarial networks during training, while increasing the resolution of the input images at the same time. They are thus able to generate realistic high-resolution images. However, their approach is not applicable to every domain, since there is no notion of resolution for some input data types, e.g. text. Sinha et al. [21] presented a strategy that blurs the activation maps of the convolutional layers using Gaussian kernel layers, reducing the noisy information caused by the network initialization. The blur level is progressively reduced to zero by decreasing the standard deviation of the Gaussian kernels. With this mechanism, they obtain a training procedure that allows the neural network to see simple information at the start of the process and more intricate details towards the end. Curriculum by Smoothing (CBS) [21] was only shown to be useful for convolutional architectures applied in the image domain. Although we found that CBS is applicable to transformers by blurring the tokens, it is not necessarily applicable to any neural architecture, e.g. standard feedforward neural networks. As an alternative to CBS, Burduja et al. [19] proposed to apply the same smoothing process on the input image instead of the activation maps. The method was applied with success in medical image alignment. However, this approach is not applicable to natural language input, as it it not clear how to apply the blurring operation on the input text.

Different from Burduja et al. [19] and Karras et al. [20], our approach is applicable to various domains, including but not limited to natural language processing, as demonstrated throughout our experiments. To the best of our knowledge, the only competing model-level curriculum method which is applicable to various domains is CBS [21]. Unlike CBS, LeRaC does not introduce new operations, such as smoothing with Gaussian kernels, during training. As such, our approach does not increase the training time with respect to the conventional training regime, as later shown in

the experiments. In summary, we consider that the simplicity of our approach comes with many important advantages: applicability to any domain and task, compatibility with any neural network architecture, time efficiency (adds no extra training time). We support all these claims through the comprehensive experiments presented in Section 4.

## 3 Method

Deep neural networks are commonly trained on a set of labeled data samples denoted as:

$$S = \{(x_i, y_i) | x_i \in X, y_i \in Y, \forall i \in \{1, 2, ..., m\}\}, \tag{1}$$

where $m$ is the number of examples, $x_i$ is a data sample and $y_i$ is the associated label. The training process of a neural network $f$ with parameters $\theta$ consists of minimizing some objective (loss) function $\mathcal{L}$ that quantifies the differences between the ground-truth labels and the predictions of the model $f$:

$$\min_{\theta} \frac{1}{m} \sum_{i=1}^{m} \mathcal{L}\left(y_i, f(x_i, \theta)\right). \tag{2}$$

The optimization is generally performed by some variant of Stochastic Gradient Descent (SGD), where the gradients are back-propagated from the neural layers closer to the output towards the neural layers closer to input through the chain rule. Let $f_1, f_2, ...., f_n$ and $\theta_1, \theta_2, ..., \theta_n$ denote the neural layers and the corresponding weights of the model $f$, such that the weights $\theta_j$ belong to the layer $f_j, \forall j \in \{1, 2, ..., n\}$. The output of the neural network for some training data sample $x_i \in X$ is formally computed as follows:

$$\hat{y}_i = f(x_i, \theta) = f_n\left(...f_2\left(f_1\left(x_i, \theta_1\right), \theta_2\right)....,\theta_n\right). \tag{3}$$

To optimize the model via SGD, the weights are updated as follows:

$$\theta_j^{(t+1)} = \theta_j^{(t)} - \eta^{(t)} \cdot \frac{\partial \mathcal{L}}{\partial \theta_j^{(t)}}, \forall j \in \{1, 2, ..., n\}, \tag{4}$$

where $t$ is the index of the current training iteration, $\eta^{(t)} > 0$ is the learning rate at iteration $t$, and the gradient of $\mathcal{L}$ with respect to $\theta_j^{(t)}$ is computed via the chain rule. Before starting the training process, the weights $\theta_j^{(0)}$ are commonly initialized with random values.

Due to the random initialization of the weights, the information propagated through the neural model during the early training iterations can contain a large amount of noise [21], which can negatively impact the learning process. Due to the feed-forward processing, we conjecture that the noise level tends to grow with each neural layer, from $f_j$ to $f_{j+1}$. The same issue can occur if the weights are pre-trained on a distinct task, where the misalignment of the weights with a new task is likely higher for the high-level feature layers. To alleviate this problem, we propose to introduce a curriculum learning strategy that assigns a different learning rate $\eta_j$ to each layer $f_j$, as follows:

$$\theta_j^{(t+1)} = \theta_j^{(t)} - \eta_j^{(t)} \cdot \frac{\partial \mathcal{L}}{\partial \theta_j^{(t)}}, \forall j \in \{1, 2, ..., n\}, \tag{5}$$

such that:

$$\eta^{(0)} \geq \eta_1^{(0)} \geq \eta_2^{(0)} \geq ... \geq \eta_n^{(0)}, \tag{6}$$

$$\eta^{(k)} = \eta_1^{(k)} = \eta_2^{(k)} = ... = \eta_n^{(k)}, \tag{7}$$

where $\eta_j^{(0)}$ are the initial learning rates and $\eta_j^{(k)}$ are the updated learning rates at iteration $k$. The condition formulated in Eq. (6) indicates that the initial learning rate $\eta_j^{(0)}$ of a neural layer $f_j$ gets lower as the level of the respective neural layer becomes higher (farther away from the input). With each training iteration $t \leq k$, the learning rates are gradually increased, until they become equal, according to Eq. (7). Thus, our curriculum learning strategy is only applied during the early training iterations, where the noise caused by the random weight initialization is most prevalent. Hence, $k$ is a hyperparameter of LeRaC that is usually adjusted such that $k \ll T$, where $T$ is the total number of

training iterations. In practice, we obtain optimal results by running LeRaC up to any epoch between 2 and 7.

We increase each learning rate $\eta_j$ from iteration 0 to iteration $k$ using an exponential scheduler that is based on the following rule:

$$\eta_j^{(l)} = \eta_j^{(0)} \cdot c^{\frac{l}{k} \cdot \left(\log_c \eta_j^{(k)} - \log_c \eta_j^{(0)}\right)}, \forall l \in \{0, 1, ..., k\}. \tag{8}$$

We set $c = 10$ in Eq. (8) across all our experiments. In practice, we obtain optimal results by initializing the lowest learning rate $\eta_n^{(0)}$ with a value that is around five or six orders of magnitude lower than $\eta^{(0)}$, while the highest learning rate $\eta_1^{(0)}$ is usually equal to $\eta^{(0)}$. Apart from these general practical notes, the exact LeRaC configuration for each neural architecture is established by tuning the hyperparameters on the available validation sets.

# 4 Experiments

## 4.1 Datasets

In general, we adopt the official data splits for the eight benchmarks considered in our experiments. When a validation set is not available, we keep $10\%$ of the training data for validation.

**CIFAR-10.** CIFAR-10 [22] is a popular dataset for object recognition in images. It consists of 60,000 color images with a resolution of $32 \times 32$ pixels. An images depicts one of 10 object classes, each class having 6,000 examples. We use the official data split with a training set of 50,000 images and a test set of 10,000 images.

**CIFAR-100.** The CIFAR-100 [22] dataset is similar to CIFAR-10, except that it has 100 classes with 600 images per class. There are 50,000 training images and 10,000 test images.

**Tiny ImageNet.** Tiny ImageNet is a subset of ImageNet [23] which provides 100,000 training images, 25,000 validation images and 25,000 test images representing objects from 200 different classes. The size of each image is $64 \times 64$ pixels.

**BoolQ.** BoolQ [24] is a question answering dataset for yes/no questions containing 15,942 examples. The questions are naturally occurring, being generated in unprompted and unconstrained settings. Each example is a triplet of the form: {question, passage, answer}. We use the data split provided in the SuperGLUE benchmark [59], containing 9,427 examples for training, 3,270 for validation and 3,245 for testing.

**QNLI.** The QNLI (Question-answering NLI) dataset [25] is a natural language inference benchmark automatically derived from SQuAD [60]. The dataset contains {question, sentence} pairs and the task is to determine whether the context sentence contains the answer to the question. The dataset is constructed on top of Wikipedia documents, each document being accompanied, on average, by 4 questions. We consider the data split provided in the GLUE benchmark [25], which comprises 104,743 examples for training, 5,463 for validation and 5,463 for testing.

**RTE.** Recognizing Textual Entailment (RTE) [25] is a natural language inference dataset containing pairs of sentences with the target label indicating if the meaning of one sentence can be inferred from the other. The training subset includes 2,490 samples, the validation set 277, and the test set 3,000 examples.

**CREMA-D.** The CREMA-D multi-modal database [27] is formed of 7,442 videos of 91 actors (48 male and 43 female) of different ethnic groups. The actors perform various emotions while uttering 12 particular sentences that evoke one of the 6 emotion categories: anger, disgust, fear, happy, neutral, and sad. Following [54], we conduct experiments only on the audio modality, dividing the set of audio samples into $70\%$ for training, $15\%$ for validation and $15\%$ for testing.

**ESC-50.** The ESC-50 [26] dataset is a collection of 2,000 samples of 5 seconds each, comprising 50 classes of various common sound events. Samples are recorded at a 44.1 kHz sampling frequency, with a single channel. In our evaluation, we employ the 5-fold cross-validation procedure, as described in related works [26, 31].

Table 1: Optimal hyperparameter settings for the various neural architectures used in our experiments.

| Architecture | Optimizer | Mini-batch | #Epochs | $\eta^{(0)}$ | CBS | | | LeRaC | |
| --- | --- | --- | --- | --- | --- | --- | --- | --- | --- |
| | | | | | $\sigma$ | $d$ | $u$ | $k$ | $\eta_1^{(0)}$ - $\eta_n^{(0)}$ |
| ResNet-18 | SGD | 64 | 100-200 | $10^{-1}$ | 1 | 0.9 | 2-5 | 5-7 | $10^{-1}$ - $10^{-8}$ |
| Wide-ResNet-50 | SGD | 64 | 100-200 | $10^{-1}$ | 1 | 0.9 | 2-5 | 5-7 | $10^{-1}$ - $10^{-8}$ |
| CvT-13 | Adamax | 64-128 | 150-200 | $2 \cdot 10^{-3}$ | 1 | 0.9 | 2-5 | 2-5 | $2 \cdot 10^{-3}$ - $2 \cdot 10^{-8}$ |
| CvT-13$_{\text{pre-trained}}$ | Adamax | 64-128 | 25 | $5 \cdot 10^{-4}$ | 1 | 0.9 | 2-5 | 3-6 | $5 \cdot 10^{-4}$ - $5 \cdot 10^{-10}$ |
| BERT$_{\text{large-uncased}}$ | Adamax | 10 | 7-25 | $5 \cdot 10^{-5}$ | 1 | 0.9 | 1 | 3 | $5 \cdot 10^{-5}$ - $5 \cdot 10^{-8}$ |
| LSTM | AdamW | 256-512 | 25-70 | $10^{-3}$ | 1 | 0.9 | 2 | 3-4 | $10^{-3}$ - $10^{-7}$ |
| SepTR | Adam | 2 | 50 | $10^{-4}$ | 0.8 | 0.9 | 1-3 | 2-5 | $10^{-4}$ - $10^{-8}$ |
| DenseNet-121 | Adam | 64 | 50 | $10^{-4}$ | 0.8 | 0.9 | 1-3 | 2-5 | $10^{-4}$ - $5 \cdot 10^{-8}$ |

## 4.2 Experimental Setup

**Architectures.** To demonstrate the compatibility of LeRaC with multiple neural architectures, we select several convolutional, recurrent and transformer models. As representative convolutional neural networks (CNNs), we opt for ResNet-18 [4], Wide-ResNet-50 [28] and DenseNet-121 [29]. As representative transformers, we consider CvT-13 [8], BERT$_{\text{uncased-large}}$ [2] and SepTr [31]. For CvT, we consider both pre-trained and randomly initialized versions. We use an uncased large pre-trained version of BERT. As Ristea et al. [31], we train SepTr from scratch. In addition, we employ a long short-term memory (LSTM) network [30] to represent recurrent neural networks (RNNs). The recurrent neural network contains two LSTM layers, each having a hidden dimension of 256 components. These layers are preceded by one embedding layer with the embedding size set to 128 elements. The output of the recurrent layers is passed to a classifier comprised of two fully connected layers. The LSTM is activated by rectified linear units (ReLU). We apply the aforementioned models on distinct input data types, considering the intended application domain of each model[1]. Hence, ResNet-18, Wide-ResNet-50 and CvT are applied on images, BERT and LSTM are applied on text, and SepTr and DenseNet-121 are applied on audio.

**Baselines.** We compare LeRaC with two baselines: the conventional training regime (which uses early stopping and reduces the learning rate on plateau) and the state-of-the-art Curriculum by Smoothing [21]. For CBS, we use the official code released by Sinha et al. [21] at `https://github.com/pairlab/CBS`, to ensure the replicability of their method in our experimental settings, which include a more diverse selection of input data types and neural architectures.

**Hyperparameter tuning.** We tune all hyperparameters on the validation set of each benchmark. In Table 1, we present the optimal hyperparameters chosen for each architecture. In addition to the standard parameters of the training process, we report the parameters that are specific for the CBS and LeRaC strategies. In the case of CBS, $\sigma$ denotes the standard deviation of the Gaussian kernel, $d$ is the decay rate for $\sigma$, and $u$ is the decay step. Regarding the parameters of LeRaC, $k$ represents the number of iterations used in Eq. (8), and $\eta_1^{(0)}$ and $\eta_n^{(0)}$ are the initial learning rates for the first and last layers of the architecture, respectively. We underline that $\eta_1^{(0)} = \eta^{(0)}$ and $c = 10$, in all experiments. Moreover, $\eta_j^{(k)} = \eta^{(0)}$, i.e. the initial learning rates of LeRaC converge to the original learning rate set for the conventional training regime. All models are trained with early stopping and the learning rate is reduced by a factor of 10 when the loss reaches a plateau.

**Evaluation.** We evaluate all models in terms of the classification accuracy. We repeat the training process of each model for 5 times and report the average accuracy and the standard deviation.

**Image preprocessing.** For the image classification experiments, we apply the same data preprocessing approach as Sinha et al. [21]. Hence, we normalize the images and maintain their original resolution, $32 \times 32$ pixels for CIFAR-10 and CIFAR-100, and $64 \times 64$ pixels for Tiny ImageNet. Similar to Sinha et al. [21], we do not employ data augmentation.

---

[1]The only exception is DenseNet-121, which is applied on audio instead of image data.

Table 2: Average accuracy rates (in %) over 5 runs on CIFAR-10, CIFAR-100 and Tiny ImageNet for various neural models based on different training regimes: conventional, CBS [21] and LeRaC. The accuracy of the best training regime in each experiment is highlighted in bold.

| Architecture | Training Regime | CIFAR-10 | CIFAR-100 | Tiny ImageNet |
|---|---|---|---|---|
| ResNet-18 | conventional | $89.20 \pm 0.43$ | $65.28 \pm 0.16$ | $57.41 \pm 0.05$ |
| ResNet-18 | CBS [21] | $89.53 \pm 0.22$ | $\mathbf{66.41} \pm 0.21$ | $55.49 \pm 0.20$ |
| ResNet-18 | LeRaC (ours) | $\mathbf{89.56} \pm 0.16$ | $66.02 \pm 0.17$ | $\mathbf{57.86} \pm 0.20$ |
| Wide-ResNet-50 | conventional | $91.22 \pm 0.24$ | $68.14 \pm 0.16$ | $55.97 \pm 0.30$ |
| Wide-ResNet-50 | CBS [21] | $89.05 \pm 1.00$ | $65.73 \pm 0.36$ | $48.30 \pm 1.53$ |
| Wide-ResNet-50 | LeRaC (ours) | $\mathbf{91.58} \pm 0.16$ | $\mathbf{69.38} \pm 0.26$ | $\mathbf{56.48} \pm 0.60$ |
| CvT-13 | conventional | $71.84 \pm 0.37$ | $41.87 \pm 0.16$ | $33.38 \pm 0.27$ |
| CvT-13 | CBS [21] | $72.64 \pm 0.29$ | $\mathbf{44.48} \pm 0.40$ | $33.56 \pm 0.36$ |
| CvT-13 | LeRaC (ours) | $\mathbf{72.90} \pm 0.28$ | $43.46 \pm 0.18$ | $\mathbf{33.95} \pm 0.28$ |
| CvT-13$_{\text{pre-trained}}$ | conventional | $93.56 \pm 0.05$ | $77.80 \pm 0.16$ | $70.71 \pm 0.35$ |
| CvT-13$_{\text{pre-trained}}$ | CBS [21] | $85.85 \pm 0.15$ | $62.35 \pm 0.48$ | $68.41 \pm 0.13$ |
| CvT-13$_{\text{pre-trained}}$ | LeRaC (ours) | $\mathbf{94.15} \pm 0.03$ | $\mathbf{78.93} \pm 0.05$ | $\mathbf{71.34} \pm 0.08$ |

Table 3: Average accuracy rates (in %) over 5 runs on BoolQ, RTE and QNLI for BERT and LSTM based on different training regimes: conventional, CBS [21] and LeRaC. The accuracy of the best training regime in each experiment is highlighted in bold.

| Architecture | Training Regime | BoolQ | RTE | QNLI |
|---|---|---|---|---|
| BERT$_{\text{large-uncased}}$ | conventional | $74.12 \pm 0.32$ | $74.48 \pm 1.36$ | $92.13 \pm 0.08$ |
| BERT$_{\text{large-uncased}}$ | CBS [21] | $74.37 \pm 1.11$ | $74.97 \pm 1.96$ | $91.47 \pm 0.22$ |
| BERT$_{\text{large-uncased}}$ | LeRaC (ours) | $\mathbf{75.55} \pm 0.66$ | $\mathbf{75.81} \pm 0.29$ | $\mathbf{92.45} \pm 0.13$ |
| LSTM | conventional | $64.40 \pm 1.37$ | $54.12 \pm 1.60$ | $59.42 \pm 0.36$ |
| LSTM | CBS [21] | $64.75 \pm 1.54$ | $54.03 \pm 0.45$ | $59.89 \pm 0.38$ |
| LSTM | LeRaC (ours) | $\mathbf{65.80} \pm 0.33$ | $\mathbf{55.71} \pm 1.04$ | $\mathbf{59.98} \pm 0.34$ |

**Text preprocessing.** For the text classification experiments with BERT, we lowercase all words and add the classification token ([CLS]) at the start of the input sequence. We add the separator token ([SEP]) to delimit sentences. For the LSTM network, we lowercase all words and replace them with indexes from vocabularies constructed from the training set. The input sequence length is limited to 512 tokens for BERT and 200 tokens for LSTM.

**Speech preprocessing.** We transform each audio sample into a time-frequency matrix by computing the discrete Short Time Fourier Transform (STFT) with $N_x$ FFT points, using a Hamming window of length $L$ and a hop size $R$. For CREMA-D, we first standardize all audio clips to a fixed dimension of $4$ seconds by padding or clipping the samples. Then, we apply the STFT with $N_x = 1024$, $R = 64$ and a window size of $L = 512$. For ESC-50, we keep the same values for $N_x$ and $L$, but we increase the hop size to $R = 128$. Next, for each STFT, we compute the square root of the magnitude and map the values to 128 Mel bins. The result is converted to a logarithmic scale and normalized to the interval $[0, 1]$. Furthermore, in all our speech classification experiments, we use the following data augmentation methods: noise perturbation, time shifting, speed perturbation, mix-up and SpecAugment [61]. The speech preprocessing steps are carried out following Ristea et al. [31].

### 4.3 Results

**Image classification.** In Table 2, we present the image classification results on CIFAR-10, CIFAR-100 and Tiny ImageNet. On the one hand, there are two scenarios (ResNet-18 on CIFAR-100 and CvT-13 on CIFAR-100) in which CBS provides the largest improvements over the conventional regime, surpassing LeRaC in the respective cases. On the other hand, there are seven scenarios where CBS degrades the accuracy with respect to the standard training regime. This shows that the improvements attained by CBS are inconsistent across models and datasets. Unlike CBS, our strategy

Table 4: Average accuracy rates (in %) over 5 runs on CREMA-D and ESC-50 for SepTr and DenseNet-121 based on different training regimes: conventional, CBS [21] and LeRaC. The accuracy of the best training regime in each experiment is highlighted in bold.

| Architecture | Training Regime | CREMA-D | ESC-50 |
|---|---|---|---|
| SepTr | conventional | $70.47 \pm 0.67$ | $91.13 \pm 0.33$ |
| SepTr | CBS [21] | $69.98 \pm 0.71$ | $91.15 \pm 0.41$ |
| SepTr | LeRaC (ours) | $\mathbf{70.95} \pm 0.56$ | $\mathbf{91.58} \pm 0.28$ |
| DenseNet-121 | conventional | $67.21 \pm 0.12$ | $88.91 \pm 0.11$ |
| DenseNet-121 | CBS [21] | $68.16 \pm 0.19$ | $88.76 \pm 0.17$ |
| DenseNet-121 | LeRaC (ours) | $\mathbf{68.99} \pm 0.08$ | $\mathbf{90.02} \pm 0.10$ |

Table 5: Average accuracy rates (in %) over 5 runs on CIFAR-10, CIFAR-100 and Tiny ImageNet for CvT-13 based on different training regimes: conventional, CBS [21], LeRaC with linear update, LeRaC with exponential update (proposed), and a combination of CBS and LeRaC.

| Architecture | Training Regime | CIFAR-10 | CIFAR-100 | Tiny ImageNet |
|---|---|---|---|---|
| CvT-13 | conventional | $71.84 \pm 0.37$ | $41.87 \pm 0.16$ | $33.38 \pm 0.27$ |
| CvT-13 | CBS [21] | $72.64 \pm 0.29$ | $44.48 \pm 0.40$ | $33.56 \pm 0.36$ |
| CvT-13 | LeRac (linear update) | $72.49 \pm 0.27$ | $43.39 \pm 0.14$ | $33.86 \pm 0.07$ |
| CvT-13 | LeRaC (exponential update) | $72.90 \pm 0.28$ | $43.46 \pm 0.18$ | $33.95 \pm 0.28$ |
| CvT-13 | CBS [21] + LeRaC | $73.25 \pm 0.19$ | $44.90 \pm 0.41$ | $34.20 \pm 0.61$ |

surpasses the baseline regime in all twelve cases, thus being more consistent. In four of these cases, the accuracy gains of LeRaC are higher than $1\%$. Moreover, LeRaC outperforms CBS in ten out of twelve cases. We thus consider that LeRaC can be regarded as a better choice than CBS, bringing consistent performance gains.

**Text classification.** In Table 3, we report the text classification results on BoolQ, RTE and QNLI. Here, there are only two cases (BERT on QNLI and LSTM on RTE) where CBS leads to performance drops compared to the conventional training regime. In all other cases, the improvements of CBS are below $0.6\%$. Just as in the image classification experiments, LeRaC brings accuracy gains for each and every model and dataset. In four out of six scenarios, the accuracy gains yielded by LeRaC are higher than $1.3\%$. Once again, LeRaC proves to be the best and most consistent regime, generally outperforming CBS by significant margins.

**Speech classification.** In Table 4, we present the results obtained on the audio data sets, namely CREMA-D and ESC-50. We observe that the CBS strategy obtains lower results compared with the baseline in two cases (SepTr on CREMA-D and DenseNet-121 on ESC-50), while our method provides superior results for each and every case. By applying LeRaC on SepTr, we set a new state-of-the-art accuracy level ($70.95\%$) on the CREMA-D audio modality, surpassing the previous state-of-the-art value attained by Ristea et al. [31] with SepTr alone. When applied on DenseNet-121, LeRaC brings performance improvements higher than $1\%$, the highest improvement ($1.78\%$) over the baseline being attained on CREMA-D.

**Additional results.** An interesting aspect worth studying is to determine if putting the CBS and LeRaC regimes together could bring further performance gains. Across all our experiments, we identified a single model (CvT-13) for which both CBS and LeRaC bring accuracy gains on all datasets (see Table 2). We thus consider this model to try out the combination of CBS and LeRaC. The corresponding results are shown in Table 5. The reported results show that the combination brings accuracy gains across all three datasets (CIFAR-10, CIFAR-100, Tiny ImageNet). We thus conclude that the combination of curriculum learning regimes is worth a try, whenever the two independent regimes boost performance.

Another important aspect is to establish if the exponential learning rate update proposed in Eq. (8) is a good choice. To test this out, we keep the CvT-13 model and change the LeRaC regime to use a linear update of the learning rate. We observe performance gains with both types of update rules,

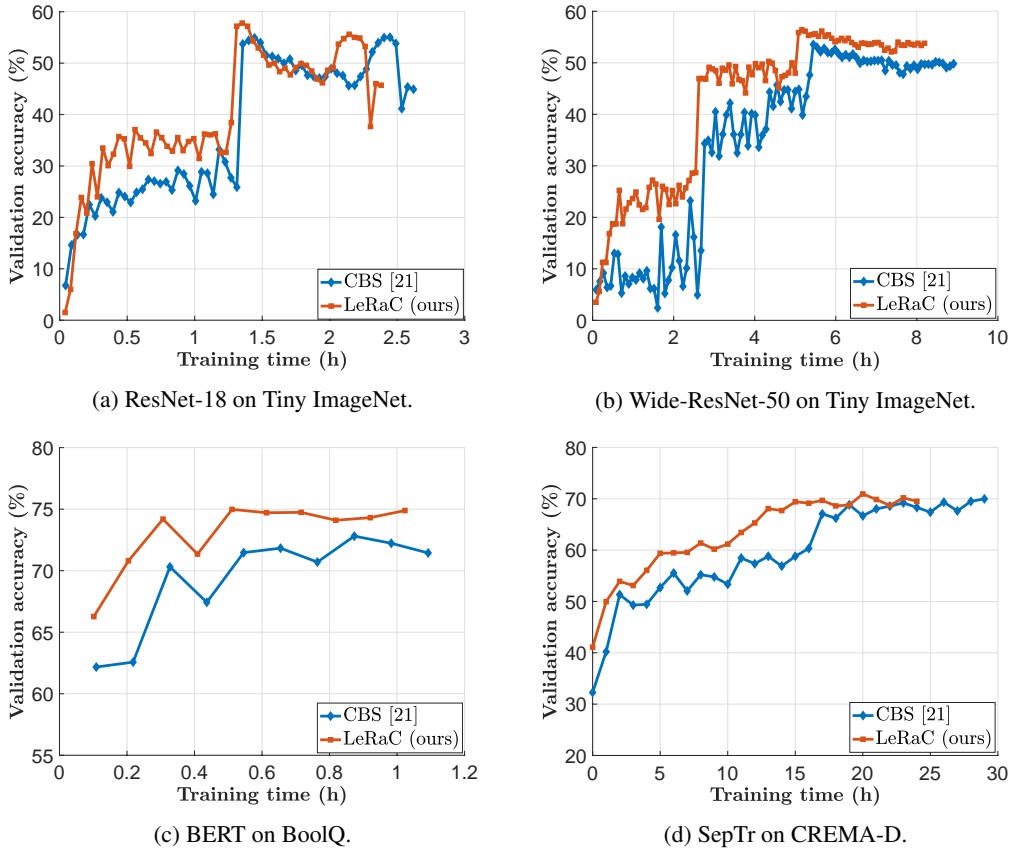

(a) ResNet-18 on Tiny ImageNet.

(b) Wide-ResNet-50 on Tiny ImageNet.

(c) BERT on BoolQ.

(d) SepTr on CREMA-D.

Figure 1: Validation accuracy (on the y-axis) versus training time (on the x-axis) for four distinct architectures. The number of training epochs is the same for both LeRaC and CBS, the observable time difference being caused by the overhead of CBS due to the Gaussian kernel layers.

but our exponential learning rate update seems to bring higher gains on all three datasets. We thus conclude that the update rule defined in Eq. (8) is a sound option.

**Training time comparison.** For a particular model and dataset, all training regimes are executed for the same number of epochs, for a fair comparison. However, the CBS strategy adds the smoothing operation at multiple levels inside the architecture, which increases the training time. To this end, we compare the training time (in hours) versus the validation error of CBS and LeRaC. For this experiment, we selected four neural models and illustrate the evolution of the validation accuracy over time in Figure 1. We underline that LeRaC introduces faster convergence times, being around 7-12% faster than CBS. It is trivial to note that LeRaC requires the same time as the conventional regime.

## 5 Conclusion

In this paper, we introduced a novel model-level curriculum learning approach that is based on starting the training process with increasingly lower learning rates per layer, as the layers get closer to the output. We conducted comprehensive experiments on eight datasets from three domains (image, text and audio), considering multiple neural architectures (CNNs, RNNs and transformers), to compare our novel training regime (LeRaC) with a state-of-the-art regime (CBS [21]) as well as the conventional training regime (based on early stopping and reduce on plateau). The empirical results demonstrate that LeRaC is significantly more consistent than CBS, perhaps being the most versatile curriculum learning strategy to date, due to its compatibility with multiple neural models and its usefulness across different domains. Remarkably, all these benefits come for free, i.e. LeRaC does not add any extra time over the conventional approach.

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
