# OpenReview forum: "LeRaC: Learning Rate Curriculum"
_NeurIPS.cc/2022/Conference — NeurIPS 2022 Submitted_

### Official Review · Reviewer_Z9wJ · 2022-07-04

**Rating:** 6
**Confidence:** 4
**Soundness:** 3 good
**Presentation:** 3 good
**Contribution:** 2 fair

**Summary:**

This paper proposes to modify neural net training by applying a different learning rate to each layer: the deeper layers are assigned a lower learning rate that progressively grows to match the first layer learning rate.

**Questions:**

== Optimization vs curriculum learning ==

The proposed method uses a different learning rate per layer. This endeavor has been widely explored in the literature on optimization of neural nets while the name curriculum learning is usually employed for methods modifying the ordering of the examples during training. I agree that the name curriculum could make sense in your context but the paper needs to spend more time on previous work on optimization. This means expanding the related work section and the empirical analysis.

The Adadelta paper (Zeiler 2012) observes the benefit to use different learning rates for different layers and Adadelta results in "effective learning rates [...] are larger for the lower layers of the network and much
smaller for the top layer at the beginning of training." Similar observations can be made for other adaptive learning rate algorithms like Adagrad, Adam, Adafactor, Adamax and AdamW or even later algorithms like e.g. SM3 (Memory-Efficient Adaptive Optimization, NeurIPS'19, see Figure 1.).

To justify your approach it seems necessary to compare your method at least with Adam and report the effective average learning rate (e.g. see Zeiler 2012's definition) per layer for both methods and how their parameters control the differentiation of learning rates across layers during training. The fact that your method is used in conjunction with Adam (or variants) could mean that either (i) the same type of learning rate schedule could be attained by changing Adam's parameters or (ii) your method's benefit comes from schedules unreachable by Adam. It seems very important to explore both methods' parameters to distinguish these two cases.

== Benefit is faster training or better generalization? ==

Modifying the training algorithm requires reporting training curves with *both* validation and training loss over time. The paper does not report any training loss which left us wonder if the generalization benefit comes from better optimization (lower training and validation loss) versus a better generalization trajectory (reaching a lower validation loss at the same training loss).

== Schedule Tuning ==

Could you explain how the exponential scheduler of Eq. 8 has been selected? Is it the only option that you tried? How is the training speed affected by the choice of the additional hyperparameters (c, \nu^{(0)}_n, k)? In particular, compared to a robust baseline like Adam, would most hyperparameter choices yield faster or slower training? I do not understand how c can be set to 10 without validation.

== Number of training steps ==

The experimental section should spend more time describing how the number of training steps was selected. In which cases the baselines would reach the same or better validation loss with more steps? E.g. Figure 1.d. seems to suggest that better validation loss with CBS can be reached with more training. This improvement does not seem to agree with the sentence "all models are trained with early stopping", does it?

== Justification of the noise argument ==

Adaptive methods (Adam etc) justify higher learning rates for first layers with a curvature argument: in laymans term, to get the same impact on the output the first layers' updates require a larger learning rates than the last ones. Adadelta (Zeiler 2012) also mentions the larger learning rate in light of the vanishing gradient effect (Bengio 2014). Your motivation (from [21]) is to reduce the effect of "noise due to untrained parameters". This argument is unclear to the reader and it would help if you could find a way to measure/characterize that "noise" and show how your method reduces its effect.


**Limitations:**

This work does not raises concerns on societal impact.


**Strengths And Weaknesses:**

The paper is clear, the approach is simple and can be implemented easily. The experiments shows better generalization when adopting the proposed learning rate schedule. The paper could be improved on two main axes:
(i) presenting the approach as an optimization method seem more appropriate than the curriculum learning angle, (ii) more analysis and justifications of why the method works seem necessary.
Overall, I feel that the paper does not explain how/if the proposed differentiated learning rate schedule is different from the effective learning rates reached by different settings of adaptive optimizers (momentum, Adadelta, Adam, SM3...).

---

> ### Author Response · Authors · 2022-08-02
> **We thank the reviewer for the positive feedback. We address the identified weaknesses in the comment below.**
>
> We thank the reviewer for the positive feedback, appreciating our clear presentation, our simple approach, as well as our empirical proof of generalization.
>
> We next address the concerns raised by the reviewer:
> - Optimization vs. curriculum learning.
> Re: We consider Adam and related optimizers as orthogonal approaches that perform the optimization itself. Our approach, LeRaC, only aims to guide the optimization during the initial training iterations by reducing the relevance of deeper network layers. As shown in Table 1, most of the baseline architectures used in our experiments are already based on Adam or some of its variations, e.g. AdaMax, AdamW. LeRaC is applied in conjunction with these optimizers, showing improved performance over various architectures and application domains. This supports our claim that LeRaC is an orthogonal contribution to the family of Adam optimizers. Nevertheless, as also suggested by reviewer mKJ3, we will add a discussion about the relationship to optimizers in our related work section. As recommended by the reviewer, we also perform a comparison between SGD+LeRaC vs. Adam. The corresponding results of ResNet18 and Wide-ResNet-50 on CIFAR-100 are shown below. We observe that our training strategy offers significantly better performance in the presented cases.
>
> ResNet18+Adam: 57.90±0.21;
> ResNet18+SGD+LeRac: 66.02±0.17.
>
> Wide-ResNet-50+Adam: 66.48±0.50;
> Wide-ResNet-50+SGD+LeRac: 69.38±0.26.
>
> - Benefit is faster training or better generalization?
> Re: As shown in Fig. 1, we achieve faster training compared to CBS. The training time of LeRaC is identical to that of the conventional regime. Compared to the conventional regime, we demonstrate better generalization via the accuracy rates reported in Tables 2, 3, 4. After analyzing our logs, we confirm that the loss values decrease for both training and validation when we introduce LeRaC. We will introduce the corresponding figures in the final paper version.
> - Schedule Tuning.
> Re: Aside from the exponential update from Eq. (8), we also tested a linear update. The corresponding results are shown in Table 5. We observe that the linear update produces slightly lower accuracy rates. Additional experiments with various hyperparameter settings are presented in our response to reviewer YyP2/oPup.
> - Why c can be set to 10 without validation.
> Re: Learning rates are usually expressed as a power of 10, e.g. 10^-4. If we start with a learning rate of 10^-8 for some layer j and we want to increase it to 10^-4 during the first k=5 epochs, the intermediate learning rates are 10^-7, 10^-6 and 10^-5, being more intuitive to understand what happens than using some other value for c. To this end, we refrained from tuning c.
> - Number of training steps.
> Re: Although we train all models with early stopping, the plots included in Fig. 1 are trained with the maximum number of epochs for both models, as we believe it provides a more fair way to compare the wall clock time of LeRaC and CBS (early stopping would introduce more variation to the time comparison). Please kindly double check Fig. 1(d) and observe that LeRaC achieves a better accuracy after 20h of training (much earlier than CBS).
> - Justification of the noise argument.
> Re: We would like to point out that the output feature maps of a layer j are affected (a) by the initial random weights (noise) θ_j^(0) of the respective layer, and (b) by the input feature maps, which are in turn affected by the random weights of the previous layers θ_1^(0), …, θ_j-1^(0). Hence, the noise affecting the feature maps increases with each layer processing the feature maps, being multiplied with the weights from each layer along the way. Our curriculum learning strategy imposes the training of the earlier layers at a faster pace, transforming the noisy weights into discriminative patterns. As noise from the earlier layer weights is eliminated, we train the later layers at faster and faster paces, until all learning rates become equal at epoch k. We will include this explanation in the camera ready.

---

### Official Review · Reviewer_mKJ3 · 2022-07-04

**Rating:** 5
**Confidence:** 4
**Soundness:** 1 poor
**Presentation:** 3 good
**Contribution:** 2 fair

**Summary:**

The paper proposes a kind of model-level curriculum learning strategy, which assigns higher initial learning rates to shallow layers than deep ones and continues increasing all learning rates until they reach the same value during the training process. It is a model- and task-agnostic approach. To verify its effectiveness, the authors apply it to multiple domains and different neural networks.

**Questions:**

1. There are many hyperparameters of this strategy, so I’d like to know how to choose proper initial learning rates for all layers and how does the sensitivity of hyperparameters affect the experimental results?
2. For pre-trained models, this paper states that “The same issue can occur if the weights are pre-trained on a distinct task, where the misalignment of the weights with a new task is likely higher for the high-level feature layers. ” However, it is generally believed that the learning rates of the layers close to the outputs are relatively high and the learning rates of the layers close to the inputs are low or even equals to zero in the process of fine-tuning, which seems to be the opposite of the strategy proposed in this paper. How to explain the difference between them?
3. It may be easier to understand Eq. (8), if the authors can plot a figure for μ(j).
4. Whether assigning higher learning rate to the deeper layers can also obtain better performance through hyper-parameter tuning ? The learning landscape of the models is not clear.

**Limitations:**

The authors do not discuss the limitation. However, there is a lack of both an intuitive explanation or theoretical analysis for the proposed method, making it hard to be a convincing work. A large line of related works is missed, making the contributions vague.

**Strengths And Weaknesses:**

Strengths:

1.The proposed strategy concerning layer-wise learning rate is important.

2.The empirical experiments are sufficient and comprehensive to prove the generality and effectiveness of the strategy.

3.This paper is well written and easy to read.

Weaknesses:

1.The relation between this work and curriculum learning is vague.  Curriculum learning is a training strategy that learns from easy to hard, or more generally, in a certain kind of meaningful order. For example, CBS, which is mentioned in the paper, anneals the standard deviation of Gaussian kernels to pass more high-frequency information. However, the relation between this work and CL is not so clear.

2.The discussed related work is not sufficient. Since this work focuses on the layer-wise learning rate, it should be discussed more clearly about how this work is different from the works about layer-wise(or adaptive) learning rate[1][2][5], which is a common and widely adopted strategy. Additionally, training with increasing learning rate in the early stages is a kind of warmup[3][4], the relations between this work and warmup strategies are required to discussed.  The vague position of this work makes its contribution vague.

3.This paper lacks theoretical analysis and does not explain why increasing the learning rates until reaching the same value is reasonable. It is not convincing only with the empirical results. The designed mechanism is neither intuitive nor well theoretically supported, making its justification questionable. The only way for justification is through experiments, which may be tricky, because the proposed method needs more hyper-parameters than that of the conventional methods.

[1] Singh, B., De, S., Zhang, Y., Goldstein, T., & Taylor, G. (2015, December). Layer-specific adaptive learning rates for deep networks. In 2015 IEEE 14th International Conference on Machine Learning and Applications (ICMLA) (pp. 364-368). IEEE.

[2] Ginsburg, B., Gitman, I., & You, Y. (2018). Large batch training of convolutional networks with layer-wise adaptive rate scaling.

[3] Gotmare, A., Keskar, N. S., Xiong, C., & Socher, R. (2018). A closer look at deep learning heuristics: Learning rate restarts, warmup and distillation.

[4] You, Y., Gitman, I., & Ginsburg, B. (2017). Large batch training of convolutional networks.

[5] Adam:A Method for Stochastic Optimization. ICLR 2015

---

> ### Author Response · Authors · 2022-08-02
> **We thank the reviewer for the relevant feedback. We address the identified weaknesses in the comment below.**
>
> The reviewer appreciates our layer-wise learning rate strategy as important, our empirical experiments as comprehensive, and our paper as well written and easy to read.
>
> Issues:
> - Relation between LeRaC and curriculum learning.
> Re: As detailed in our reply to reviewer YyP2, our method can be seen as a curriculum learning strategy that simplifies the optimization in the early training stages by restricting the model updates (in a soft manner) to certain directions (corresponding to the weights of the earlier layers). Due to the imposed soft restrictions (lower learning rates for deeper layers), the optimization is easier at the beginning. As the training progresses, all directions become equally important, and the loss function is permitted to be optimized in any direction. As the number of directions grows, the optimization task becomes more complex (it is harder to find the optimum). The relationship to curriculum learning can be discovered by noting that the complexity of the optimization increases over time, just as in curriculum learning.
> - Extend related work.
> Re: We thank the reviewer for pointing out related work on adaptive learning rates and warmup strategies (we relabel the indicated works with [R1,R2,R3,R4,R5] to avoid confusion). We agree that [R1-R5] are related to our work and we explain the differences below. In [R1], the main goal is increasing the learning rate of certain layers as necessary, to escape saddle points. Different from [R1], our strategy reduces the learning rates of deeper layers, introducing soft optimization restrictions in the initial training epochs. In [R2,R4], the authors proposed to train models with very large batches using a learning rate for each layer, by scaling the learning rate with respect to the norms of the weights / gradients. The goal of [R2,R4] is to specifically learn models with large batch sizes, e.g. formed of 8K samples. Unlike [R2,R4], we propose a more generic approach that can be applied to multiple architectures (convolutional, recurrent, transformer) under unrestricted training settings. In [R3], the authors point out that learning rate with warmup restarts is an effective strategy to improve stability of training neural models using large batches. Different from LeRaC, this approach does not employ a different learning rate for each layer. Moreover, the strategy restarts the learning rate at different moments during the entire training process, while LeRaC is applied only during the first few training epochs. Aside from these technical differences, our experiments already include a comparison of the two strategies (LeRaC vs. Linear Warmup with Cosine Annealing), as also pointed out in the response to reviewer YyP2. Indeed, the CvT results presented in Table 2 show that introducing LeRaC brings consistent improvements. We thus conclude that our strategy is a viable and distinct alternative to learning rate warmup. The relationship to Adam [R5] is detailed in the reply to reviewer Z9wJ.
> - Paper lacks theoretical analysis.
> Re: We explained the intuition behind our approach in answer to the first point. We would like to add that many contributions in the area of machine learning are not always supported by theoretical proof, and are only proven to work empirically. In some cases, the theory is developed after the empirical evidence is observed. We strongly believe that empirical studies backed by a good intuition offer a path forward that can raise the interest of many researchers.
> - Choosing initial learning rates for all layers. Sensitivity of hyperparameters.
> Re: In general, we note that our hyperparameters are tuned on validation data. Upon setting the range for the initial learning rates, i.e. η_1^(0) and η_n^(0), we set the intermediate initial learning rates using a similar formula to Eq. (8), the main difference being that initial learning rates variate according to the layer j instead of the training iteration l. To observe the sensitivity of the results to hyperparameters, we kindly ask the reviewer to see the reply to reviewers YyP2/oPup.
> - Higher learning rate to deeper layers.
> Re: We refer to this strategy as “anti-curriculum”. We present some results in the reply to reviewer YyP2.
> - For pre-trained models, fine-tuning should use the opposite strategy.
> Re: Our principle is to first let the generic (low-level) features get adapted to the new task, then optimize the deeper (high-level) features to the new task. If the generic features do not need to be adapted, their gradients will be small and the use of the usual learning rate will not affect the model. However, we do not argue against using “anti-curriculum” [12], which worked well in multiple cases. We agree with the general perspective, which we believe is valid for the entire duration of the training process. We note that our principle is only applied for the first few training iterations.
> - Figure for Eq. (8). Re: We will add it.
> - Limitations. Re: Please see reply to reviewer oPup.

---

> > ### Comment · Reviewer_mKJ3 · 2022-08-03
> > **Reply to the rebuttal**
> >
> > Thank the authors' effort for the rebuttal. The author addresses some of my concerns like the differences between this work and the given related works, and also, some conflicts with pretrain-fintune paradigm. Despite of the addressed concerns, there are still some important points in this paper which are still vague, like whether the noise of the shallow layers is larger in the beginning, which might be shown by the distance to the final model?  Additionally, since the initial learning rate for each layer and the number of curriculum epoch needs tuning, this work seems like to introduce more hyper-parameters to achieve promising performance and the role of curriculum learning seems to become less ? The most confusing part of this paper is still how this method work, although it achieves promising performance on various scenarios. Out of all these considerations, I think this is a borderline paper that may bring performance improvement through tuning the proposed method, but the mechanism of this work needs more explanation and exploration. I will increase my score to 4.

---

> > > ### Author Response · Authors · 2022-08-03
> > > **We thank the reviewer for increasing the score. We address remaining concerns below.**
> > >
> > > We thank the reviewer for reading our rebuttal and increasing their score based on the concerns that were addressed in the rebuttal. The reviewer still considers that there are some important points that require further attention. We address these concerns below:
> > >
> > > - Quantify noise of shallow layers using distance to the final model.
> > > Re: In our reply to reviewer YyP2 on training dynamics, we computed the entropy of the low-level and high-level layers after k=6 epochs, before and after using LeRaC to train ResNet-18 on CIFAR-10. We agree that the idea of using the distance to final feature maps provides additional useful insights about how LeRaC works. To this end, we computed the Euclidean distances of the low-level features between epoch k and the final epoch, before and after using LeRaC. We did the same for the high-level features. The distances are shown below. The computed distances confirm our previous observations: LeRaC seems to balance the training pace of low-level and high-level layers.
> > >
> > > ResNet-18 {format is (1st layer distance, last layer distance)}:
> > > (0.60, 0.37).
> > >
> > > ResNet-18+LeRaC:
> > > (0.61, 0.66).
> > >
> > > - The work seems to introduce more hyperparameters.
> > > Re: It is true that LeRaC adds three additional hyperparameters compared to the conventional training regime. These are the initial highest and lowest learning rates, η_1^(0) and η_n^(0), and the number of iterations k to employ LeRaC. We tried our best to minimize the number of hyperparameters that require tuning by using fixed rules to adjust intermediate learning rates (e.g. Eq. (8)) or by fixing less important hyperparameters, e.g. c=10. As shown in Table 1, CBS has an identical number of additional hyperparameters to LeRaC. Furthermore, we note that data-level curriculum methods also introduce additional hyperparameters. Even a simple method that splits the examples into easy-to-hard batches that are gradually added into the training set requires at least two hyperparameters: the number of batches (should we use 3 batches - easy, medium and hard - or more?), and the number of iterations before introducing a new training batch. We thus believe that, in terms of the number of additional hyperparameters, LeRaC is comparable to CBS and other curriculum learning strategies. Please note that the same happens if we look at optimizers, e.g. Adam adds two additional hyperparameters compared to SGD.
> > >
> > > - Confusion on how the method works.
> > > Re: From a technical point of view, we note that our approach can also be regarded as a way to guide the optimization, which we see as an alternative to loss function smoothing. The link between curriculum learning and loss smoothing is mentioned in [12], where the authors suggest that curriculum learning strategies induce a smoothing of the loss function, where the smoothing is higher during the early training iterations (simplifying the optimization) and lower to non-existent during the late training iterations (restoring the complexity of the loss function). LeRaC is aimed at producing a similar effect, but in a softer manner by dampening the importance of optimizing the weights of high-level layers in the early training iterations. Additionally, please note that considering the reply to reviewer YyP2 on training dynamics, we observe that LeRaC tends to balance the training pace of low-level and high-level features, while the conventional regime seems to update the high-level layers at a faster pace. This could provide an additional intuitive explanation of why our method works. We will include our additional comments and results in the final paper to improve clarity.

---

> > > > ### Comment · Reviewer_mKJ3 · 2022-08-04
> > > > **Reply to the author**
> > > >
> > > > Thank the authors for the reply again. The distances shown should be the initial weights compared to the final weights, because the authors state that the shallow layers contain less noise. It will be better to show the distances changes with layers.

---

> > > > > ### Author Response · Authors · 2022-08-04
> > > > > **Sorry for the misunderstanding.**
> > > > >
> > > > > We thank the reviewer for their patience and we apologize for misunderstanding the issue regarding the distances. We have now computed the distances for the low-level (first conv) and high-level (last conv) layers between the values at iteration 0 (based on random values) and the last iteration (based on values optimized until convergence) for ResNet-18 on CIFAR-10 (same model as before). The computed distances (shown below) confirm our conjecture: shallow layers contain less noise than deep layer. We hope our current answer can satisfy the reviewer's concern.
> > > > >
> > > > > ResNet-18: (low-level distance = 38.36, high level-distance = 709.93).

---

> > > > > > ### Comment · Reviewer_mKJ3 · 2022-08-04
> > > > > > **Reply to the rebuttal**
> > > > > >
> > > > > > Thanks for the additional experiment again. Considering the satisfying answers from the authors, I'd like to increase my score to 5.

---

> > > > > > > ### Author Response · Authors · 2022-08-04
> > > > > > > **We thank the reviewer once more!**
> > > > > > >
> > > > > > > We once again thank the reviewer for reading and acknowledging all our clarifications and for deciding to further increase the rating. We do believe the discussion was very fruitful, which will lead to significant improvements of our final manuscript.

---

### Official Review · Reviewer_oPup · 2022-07-10

**Rating:** 5
**Confidence:** 3
**Soundness:** 4 excellent
**Presentation:** 4 excellent
**Contribution:** 3 good

**Summary:**

The manuscript proposes a model-level curriculum learning framework that uses variable learning rates across a DNN to jump-start the learning process. The core idea revolves around compensating for the difference in learning across the depth of a DNN by using larger learning rates closer to the input and gradually decreasing their value towards the output layers. As intended, this curriculum is data-free and removes the need to assess the difficulty of samples, similar to how data-level curriculum operates, and instead can be applied across a range of architectures and tasks. The proposed idea improves on the performance of standard training regimes across vision, language and audio tasks, using residual, recurrent and transformer-based architectures.

**Questions:**

- Specific to the vision datasets and results, could the authors provide insights into (a) how the use of common preprocessing techniques affects the behavior of the proposed approach, and (b) the balance in peak performance achievable in standard training vs. the proposed method when preprocessing is included. While preprocessing techniques add some element of stochasticity, it is important to understand an algorithm in that context.
- The addition of significance tests/metrics would be beneficial to further emphasize the benefits of the proposed algorithm as well as other areas for potential improvement. Could the authors reassess their results using significance metrics?
- Moving a small portion of the experimental setup to the appendices/supplementary material and including a study of how k (iterations/epochs), from small to absurdly large, affects the improvement in performance offered by the proposed algorithm would add another dimension to the evaluations presented in this work. The results themselves should be a by-product of the suite of experiments already executed so hopefully there is minimal overhead.
- Given the variable levels of improvement, even within the scope of the same task, could the authors discuss plausible schemes that take into account other factors (like data, gradient flow, etc.) that could provide more consistent gains?
- At a more abstract level could the authors address the potential negative implications/limitations of their approach.

**Limitations:**

While there aren't glaring limitations or issues with the proposed method, the authors can consider potential negative effects in alternative properties of the DNN, as a by-product.
Since there isn't an evaluation of the calibration quality or adversarial robustness of the final solutions, these could be potential issues or limitations to the proposed algorithm.
Keeping robust/adversarial training or alternative objective functions to classification in mind could lead to unexpected outcomes.


**Strengths And Weaknesses:**

Strengths
- The writing offers a clear and understandable representation of the curriculum learning approach.  More importantly, the categorization of various levels of curriculum learning provides important context to the ideas being discussed.
- The depth of experimental evaluation, stretching across datasets, tasks and architectures is commendable.

Weaknesses
- Data augmentation has proven valuable to the training process, be it commonly used approaches like random cropping/flipping or more intricate mixed approaches. They offer substantial improvement in performance and as well as alternative properties of a DNN. The proposed method is not evaluated alongside any common preprocessing techniques (vision datasets), thus offering a gap in the exploration of the behavior of the proposed algorithm alongside them.
- The set of results posted across various tables in the manuscript is commendable. However, in certain cases the the average performance and standard deviation across multiple methods are extremely close. While this does not take away from the final conclusions, the use of significance tests/metrics could help further distinguish improvements.
- The descriptions of the preprocessing and training regimes offer extremely important context to the results presented throughout the paper.
However, an exploration of the two key parameters that make up the proposed method, various update setting for the learning rate and the choice of iteration/epoch at which to converge to the standard learning rate would offer more insight into the proposed method.

---

> ### Author Response · Authors · 2022-08-02
> **We thank the reviewer for the positive feedback. We address the identified issues in the comment below.**
>
> We thank the reviewer for the positive feedback, appreciating our clear and understandable presentation of curriculum learning, and our comprehensive experimental evaluation.
>
> We next address the identified issues:
> - Not using augmentation on vision data offers a gap in exploring the behavior of LeRaC.
> Re: Following [21], we did not use data augmentation for the vision datasets. We consider data augmentation as an orthogonal method for improving results, expecting improvements for both baseline and LeRaC models. Furthermore, since we extended the experimental settings to other domains, we took the liberty to use data augmentation in the audio domain. The same augmentations (noise perturbation, time shifting, speed perturbation, mix-up and SpecAugment [61]) are used for all audio models, ensuring a fair comparison. As the reviewer suggested, we present new results with ResNet-18 and Wide-ResNet-50 on CIFAR-100 using the following augmentations: horizontal flip, rotation, solarize, blur, sharpness, auto-contrast. The results confirm that the performance gaps in the vision domain are in the same range after introducing data augmentation. We will add the extra results in the final version.
>
> ResNet-18+data augmentation:
> (baseline: 71.25±0.04);
> (LeRaC: 71.52±0.22).
>
> Wide-ResNet-50+data augmentation:
> (baseline: 65.42±0.66);
> (LeRaC: 67.00±0.55).
>
> - Significance testing.
> Re: As the reviewer suggested, we applied McNemar significance testing to determine if the differences with respect to the baseline are significant. In 17 of 22 cases, we found that our results are significantly better than the corresponding baseline, at a confidence threshold of 0.001. We will mark the significantly better results in the final paper version.
>
> - Exploration of the two key parameters.
> Re: In general, we note that our hyperparameters are tuned on the validation data. As the reviewer suggested, we present additional results with different ranges for η_1^(0) and  η_n^(0) in the response to reviewer YyP2. Below, we also present results with ResNet-18 and Wide-ResNet-50 on CIFAR-100, considering various values for k. We observe that all configurations surpass the baselines on CIFAR-100. Moreover, we observe that the optimal values for k (k=7 for ResNet-18 and k=7 for Wide-ResNet-50) obtained on the validation set are not the values producing the best results on the test set. We will add these new results in the final ablation study.
>
> ResNet-18+LeRac {format is (k,acc)}:
> (5, 66.25±0.07);
> (6, 66.20±0.05);
> (7, 66.02±0.17);
> (8, 66.91±0.23);
> (9, 66.59±0.47).
>
> Wide-ResNet-50+LeRac:
> (5, 68.86±0.76);
> (6, 69.78±0.16);
> (7, 69.38±0.26);
> (8, 69.30±0.18).
>
> - Discuss plausible schemes taking into account other factors (like data, gradient flow, etc.) that could provide more consistent gains?
> Re: Our aim was to propose a simple and generic curriculum learning scheme, which can be integrated into any model for any task. To this end, we tried to avoid relying on task dependent information (e.g. data). In Table 5, we showed that combining LeRaC and CBS can boost performance. In a similar fashion, LeRaC can be combined with data-level curriculum strategies for improved performance. We leave this exploration for future work. Further performance gains can be obtained by introducing orthogonal approaches, e.g. data augmentation.
>
> - Address the potential negative implications / limitations.
> Re: One limitation, indicated by reviewer YyP2, is the need to disable other learning rate schedulers while using LeRaC. We already tested this scenario with CvT (the baseline CvT uses Linear Warmup with Cosine Annealing, which is removed when using LeRaC), observing performance gains (see Table 2 form the paper). However, disabling alternative learning rate schedulers might bring performance drops. Hence, this has to be decided on a case by case basis. Another limitation is longer training times / poor convergence if the hyperparameters are not properly configured. We recommend hyperparameter tuning on validation to avoid this outcome. Another limitation is that we tested our approach on mainstream classification tasks involving mainstream classification losses (multi-class / binary cross-entropy). We leave the integration with additional losses for future work.

---

> > ### Comment · Reviewer_oPup · 2022-08-04
> > **Response to rebuttal**
> >
> > I appreciate the authors' detailed response to my comments. I would like to add a few additional comments based on the responses provided in the rebuttal.
> > - The additional evaluations on vision data using augmentations helps highlight how much the proposed work can improve on stronger baselines. As currently provided, the additional evaluations and their relevant significance tests should help showcase the potential of the proposed method. I would encourage the authors to explore the best possible performance for the chosen Dataset-DNN combination and push to improve over it.
> > - It is indeed interesting that across ablation studies on k, the proposed method consistently performs better than the baseline methods while the impact of varying $\eta_1^{0}$ is much more significant, possibly indicating a weaker threshold on k and allowing it to be freely set. Could the authors confirm whether modifying the relative distance between the learning rates of each layer leads to the same impact, as opposed to the selection of specific values, i.e., the $\delta$ between learning rates of layers is more critical than the exact choice of value? The reason this could be integral is, if the difference is key then the idea can be adopted across any/all setups without having to re-evaluate specific new learning rates.

---

> > > ### Author Response · Authors · 2022-08-05
> > > **We thank the reviewer reading our rebuttal. We address the additional concerns below.**
> > >
> > > We thank the reviewer for taking the time to read our rebuttal. We address the additional concerns below:
> > > - Explore the best possible performance for the chosen Dataset-DNN combination and push to improve over it.
> > > Re: We thank the reviewer for this suggestion. We will use it to improve our results in the final paper version.
> > > - Is modifying the relative distance between the learning rates sufficient?
> > > Re: First, we would like to underline that the delta between the initial learning rates of consecutive layers is auto-set based on the range η_1^(0) and η_n^(0). For example, let us consider a network with 5 layers. If we choose η_1^(0)=10^-1 and η_5^(0)=10^-2, than the intermediate initial learning rates are η_2^(0)=10^-1.25, η_3^(0)=10^-1.5, η_4^(0)=10^-1.75, i.e. delta is used in the exponent and is equal to -0.25 in this case. Please note that, to obtain the intermediate learning rates, we use an exponential scheduler (similar to Eq. (8)), as mentioned in the rebuttal to reviewer mKJ3. In general, we underline that it is sufficient to set two of the three hyperparameters (η_1^(0), η_n^(0), delta), as the third one can be directly inferred from the other two. In our case, we opted to infer delta from η_1^(0) and η_n^(0). Based on our understanding, the reviewer is asking if it is sufficient to set delta. This is basically equivalent to keeping the difference between η_1^(0) and η_n^(0) fixed, while changing only η_1^(0). To test this scenario, we perform experiments with Wide-ResNet-50 on CIFAR-100 and present the results below. The results indicate that keeping delta fixed while changing η_1^(0) can lead to different accuracy rates. Hence, we conclude that tuning at least two of the three hyperparameters (η_1^(0), η_n^(0), delta) is necessary.
> > >
> > > Wide-ResNet-50+LeRac on CIFAR-100 {format is (η_1^(0), η_n^(0), acc)}:
> > > (10^-1, 10^-7, 69.25±0.37);
> > > (10^-2, 10^-8, 68.51±0.52);
> > > (10^-3, 10^-9, 68.38±0.06).

---

### Official Review · Reviewer_YyP2 · 2022-07-23

**Rating:** 6
**Confidence:** 4
**Soundness:** 2 fair
**Presentation:** 4 excellent
**Contribution:** 3 good

**Summary:**

This paper proposes setting different learning rates for each layer of a deep neural network such that the first layers (closer to the input) have a much higher learning rate than the last layers. Learning rates are all gradually increased until they reach the same value $\eta_0$ (the optimal learning rate of no-curriculum models).

**Questions:**

- What is the method for optimizing $\eta_j$, j = {1 ... n}?
- How sensitive is the model to different settings of $\eta_j$?
- How do the training dynamics compare to standard training where all layers are trained at the same rate? Would the earlier layers generate qualitatively different features maps than under-trained later layers?

**Limitations:**

- The interaction of the model with adaptive optimizers that scale the learning rate of each parameter is not well justified.
- Learning rate schedulers may disturb the proposed model. If the base learning rate is reduced before the learning rate of the last layer $\eta_n$ is sufficiently increased, the last layer may never be trained enough, training may take much longer or may become unstable.

**Strengths And Weaknesses:**

Strengths:
- Simple, novel, and effective method.
- Good review of related work and reasoning for using model-level curriculum learning.
- Consistent performance improvements across eight audio, image, and text datasets and seven convolutional, recurrent, and transformer architectures.


Weaknesses:
- Insufficient analysis on why the method works.
- The motivation for setting different learning rates per layer is not sound. The authors state that random initialization of parameters causes the propagation of noise during the forward pass, and that assigning higher learning rate to first layers and lower learning rates to last layers prevents this propagation of noise. But there is no proof for this argument.

---

> ### Author Response · Authors · 2022-08-02
> **We thank the reviewer for the generally positive feedback. We address the identified issues in the comment below.**
>
> The reviewer appreciates our simple, novel, and effective method, our good literature review, and our consistent performance improvements over multiple architectures and datasets.
>
> Issues:
> - Motivation for setting different learning rates per layer.
> Re: Note that the output feature maps of a layer j are affected by (a) the initial random weights (noise) θ_j^(0) of the respective layer, and by (b) the input feature maps, which are in turn affected by the random weights of the previous layers θ_1^(0),…,θ_j-1^(0). Hence, the noise affecting the feature maps increases with each layer processing the feature maps, being multiplied with the weights from each layer along the way. Our curriculum learning strategy imposes the training of the earlier layers at a faster pace, transforming the noisy weights into discriminative patterns. As noise from the earlier layer weights is eliminated, we train the later layers at faster and faster paces, until all learning rates become equal at epoch k. We will include this explanation in the camera ready.
>
> - Optimization method for η_j, j={1...n}.
> Re: Note that the different learning rates η_j are not optimized during training. We set the initial learning rates η_j^(0) through validation, such that η_n^(0) is around five or six orders of magnitude lower than η^(0) and η_1^(0)=η^(0). After initialization, we apply the scheduler defined in Eq. (8).
>
> - Model sensitivity to different settings of η_j.
> Re: Since our goal was to perform curriculum learning, we restricted the settings for η_j according to Eqs. (6) and (7). As also suggested by reviewer mKJ3, another strategy is to consider the opposite setting, where we use higher learning rates for the deeper layer. We tested this approach, which we call “anti-curriculum”, in a set of new experiments with ResNet-18, Wide-ResNet-50 on CIFAR-100 and SepTr on CREMA-D. The results are shown below. Although anti-curriculum, e.g. hard negative sample mining, was shown to be useful in other tasks [12], our results indicate that learning rate anti-curriculum attains inferior performance. In another set of experiments, we show results with LeRaC using different ranges for η_1^(0) and η_n^(0). We observe that there are multiple hyperparameter configurations that surpass the baseline.
>
> Anti-curriculum:
> ResNet-18 on CIFAR-100: 64.76±0.17;
> Wide-ResNet-50 on CIFAR-100: 67.47±0.15;
> SepTr on CREMA-D: 68.33±0.61.
>
> ResNet-18+LeRaC on CIFAR-100 {format is (η_1^(0),η_n^(0),acc)}:
> (10^-1,10^-6,65.82±0.08);
> (10^-1,10^-7,65.80±0.16);
> (10^-1,10^-9,65.59±0.49);
> (10^-1,10^-10,65.76±0.22);
> (10^-2,10^-8,65.71±0.08);
> (10^-3,10^-8,65.25±0.12).
>
> Wide-ResNet-50+LeRac on CIFAR-100:
> (10^-1,10^-6,68.64±0.52);
> (10^-1,10^-7,69.25±0.37);
> (10^-1,10^-9,69.26±0.27);
> (10^-1,10^-10,69.66±0.34);
> (10^-2,10^-8,68.51±0.52);
> (10^-3,10^-8,68.71±0.47).
>
> SepTr+LeRaC on CREMA-D:
> (10^-2,10^-8,70.74±0.55);
> (10^-3,10^-8,70.61±0.49);
> (10^-5,10^-8,70.32±0.57);
> (10^-4,10^-7,70.49±0.44);
> (10^-4,10^-9,70.58±0.48).
>
> - Training dynamics. Quality of features maps of earlier layers vs. under-trained later layers.
> Re: We showed a few examples of training dynamics in Fig. 1. All four graphs exhibit a higher gap between standard training and LeRaC in the first half of the training process, suggesting that LeRaC has an important role towards faster convergence. To assess the comparative quality of low-level vs. high-level features maps obtained with conventional vs. LeRaC training, we compute the entropy of the first and last conv layers of ResNet-18 on CIFAR-10 after k=6 iterations. Conventional training seems to update deeper layers faster, observing a higher difference between the entropies of low-level and high-level features obtained with conventional training than with LeRac. Hence, LeRaC balances the training pace of low-level and high-level features.
>
> ResNet-18 {format is (1st layer entropy, last layer entropy)}:
> (0.99646, 0.99050).
> ResNet-18+LeRaC:
> (0.99706, 0.99683).
>
> - Interaction of LeRaC with adaptive optimizers.
> Re: Our initial learning rates and scheduler are used independently of the optimizers. As shown in Table 1, we used different optimizers, opting in each case for the best optimizer for each baseline architecture (without LeRaC). We kept the same optimizers when introducing LeRaC. Our learning rate scheduler updates the learning rates at the beginning of every iteration. We did not observe any stability / interaction issues.
>
> - Learning rate schedulers may disturb the proposed model.
> Re: Whenever a learning rate scheduler was used for training a model, we simply replaced the scheduler with LeRaC. For example, all the baseline CvT results are based on Linear Warmup with Cosine Annealing (LWCA), this being the recommended scheduler for CvT. When we introduced LeRaC, we simply deactivated LWCA. In general, we recommend deactivating other schedulers while using LeRaC for simplicity in avoiding stability issues. We will mention this limitation in the final version of the paper.

---

### Meta-Review · Area_Chair_yMWK · 2022-08-31

**Recommendation:** Reject
**Confidence:** Less certain

**Metareview:**

The paper proposes a model-level curriculum learning strategy, which assigns higher initial learning rates to shallow layers than deep ones and continues increasing all learning rates until they reach the same value during the training process. It is a model- and task-agnostic approach.

Reviewers appreciated the simplicity of the approach, as well as its effectiveness on a multiple domains and different neural networks.

Main concerns which remains after the rebuttal are:
- There is an insufficient analysis on why the method works. Some intuition has been given in the rebuttal, but all reviewers felt more analysis should be given to reach NeurIPS standards.
- There is insufficient comparison with previous work on optimization.

In addition, a more minor concern (in AC's opinion) remains:
- On vision data, evaluations using augmentations are missing. While it can be argued that the effect of augmentations and LeRaC might be orthogonal, it remains unclear how one can improve on strong baselines there.


**Award:**

No

---

### Decision · Program_Chairs · 2022-09-14

Reject

---

> ### Public Comment · Authors · 2022-11-18
> **Authors' opinion on final decision (continued)**
>
> We provided the following answer to reviewer Z9wJ: "We consider Adam and related optimizers as orthogonal approaches that perform the optimization itself. Our approach, LeRaC, only aims to guide the optimization during the initial training iterations by reducing the relevance of deeper network layers. As shown in Table 1, most of the baseline architectures used in our experiments are already based on Adam or some of its variations, e.g. AdaMax, AdamW. LeRaC is applied in conjunction with these optimizers, showing improved performance over various architectures and application domains. This supports our claim that LeRaC is an orthogonal contribution to the family of Adam optimizers. Nevertheless, as also suggested by reviewer mKJ3, we will add a discussion about the relationship to optimizers in our related work section. As recommended by the reviewer, we also perform a comparison between SGD+LeRaC vs. Adam. The corresponding results of ResNet18 and Wide-ResNet-50 on CIFAR-100 are shown below. We observe that our training strategy offers significantly better performance in the presented cases. ResNet18+Adam: 57.90±0.21; ResNet18+SGD+LeRac: 66.02±0.17. Wide-ResNet-50+Adam: 66.48±0.50; Wide-ResNet-50+SGD+LeRac: 69.38±0.26."
>
> Reviewer Z9wJ did not further acknowledge reading our rebuttal.
>
> Overall, we consider that this point was addressed in the rebuttal, as acknowledged by reviewer mKJ3 (the only one who read the rebuttal among the three).
>
> > Area Chair yMWK: "On vision data, evaluations using augmentations are missing. While it can be argued that the effect of augmentations and LeRaC might be orthogonal, it remains unclear how one can improve on strong baselines there."
>
> Reply: Concerns regarding data augmentation were raised by reviewer oPup. We provided the following answer to the reviewer: "Following [21], we did not use data augmentation for the vision datasets. We consider data augmentation as an orthogonal method for improving results, expecting improvements for both baseline and LeRaC models. Furthermore, since we extended the experimental settings to other domains, we took the liberty to use data augmentation in the audio domain. The same augmentations (noise perturbation, time shifting, speed perturbation, mix-up and SpecAugment [61]) are used for all audio models, ensuring a fair comparison. As the reviewer suggested, we present new results with ResNet-18 and Wide-ResNet-50 on CIFAR-100 using the following augmentations: horizontal flip, rotation, solarize, blur, sharpness, auto-contrast. The results confirm that the performance gaps in the vision domain are in the same range after introducing data augmentation. We will add the extra results in the final version. ResNet-18+data augmentation: (baseline: 71.25±0.04); (LeRaC: 71.52±0.22). Wide-ResNet-50+data augmentation: (baseline: 65.42±0.66); (LeRaC: 67.00±0.55)."
>
> Reviewer oPup was happy with our clarification, stating that "The additional evaluations on vision data using augmentations helps highlight how much the proposed work can improve on stronger baselines. As currently provided, the additional evaluations and their relevant significance tests should help showcase the potential of the proposed method."
>
> We thus consider that this point was addressed in the rebuttal, as acknowledged by reviewer oPup.
>
> In conclusion, we believe that Area Chair yMWK did not take all the rebuttal comments into consideration.
> On September 15th, we e-mailed the General and Program Chairs and kindly asked them to reassess our case, as we believe it was mishandled by Area Chair yMWK. The content of the e-mail is the same as the comments above. We did not receive a reply. We also sent a reminder with the following content:
>
> "Over 9 days have passed since we reported how poorly our manuscript was handled by the allocated Area Chair at NeurIPS 2022. As we did not receive a reply until now, it becomes clear that our case is simply going to be ignored. While we are fairly confident that this is not how such cases should be treated, especially at top-tier conferences, we feel deeply sorry to find out that we are beneath your esteemed selves and that we do not deserve a reply. We surely hope that the way you treated us, with ignorance, will not become the standard at future NeurIPS or other conferences where you are chairs, reviewers or authors. Otherwise, the publishing system is likely going to take a very bad turn."
>
> On November 18th, we still did not receive a reply or clarification.

---

> ### Public Comment · Authors · 2022-11-18
> **Authors' opinion on final decision**
>
> We believe that our Area Chair yMWK did not carefully consider the rebuttal comments and clearly overlooked the final positive recommendations (**2 x weak accept, 2 x borderline accept**) of the four reviewers. More specifically, based on their ratings, all four reviewers believe that the positive points outweigh the negative ones. Moreover, the Area Chair recognizes that their "Confidence" is "Less certain".
>
> The Area Chair's reject decision is based on the following three concerns, which we address below:
>
> > Area Chair yMWK: "There is an insufficient analysis on why the method works. Some intuition has been given in the rebuttal, but all reviewers felt more analysis should be given to reach NeurIPS standards."
>
> Reply: Regarding this point, we note that two reviewers (mKJ3 and oPup) interacted during the discussion period. Neither of them seemed concerned about the lack of analysis after seeing our rebuttal comments. In fact, reviewer mKJ3 turned his recommendation to the positive side, which indicates that they were pleased with our clarifications. The other two reviewers (YyP2 and Z9wJ) did not acknowledge reading our rebuttal at all. None of the reviewers mentioned that the "analysis is below NeurIPS standards". We agree that "all reviewers felt more analysis should be given", but only before the rebuttal. However, this was not the case after the rebuttal and discussion period ended.
>
> > Area Chair yMWK: "There is insufficient comparison with previous work on optimization."
>
> Reply: This concern was raised by reviewers mKJ3, YyP2 and Z9wJ.
>
> We provided the following answer to reviewer mKJ3: "We thank the reviewer for pointing out related work on adaptive learning rates and warmup strategies (we relabel the indicated works with [R1,R2,R3,R4,R5] to avoid confusion). We agree that [R1-R5] are related to our work and we explain the differences below. In [R1], the main goal is increasing the learning rate of certain layers as necessary, to escape saddle points. Different from [R1], our strategy reduces the learning rates of deeper layers, introducing soft optimization restrictions in the initial training epochs. In [R2,R4], the authors proposed to train models with very large batches using a learning rate for each layer, by scaling the learning rate with respect to the norms of the weights / gradients. The goal of [R2,R4] is to specifically learn models with large batch sizes, e.g. formed of 8K samples. Unlike [R2,R4], we propose a more generic approach that can be applied to multiple architectures (convolutional, recurrent, transformer) under unrestricted training settings. In [R3], the authors point out that learning rate with warmup restarts is an effective strategy to improve stability of training neural models using large batches. Different from LeRaC, this approach does not employ a different learning rate for each layer. Moreover, the strategy restarts the learning rate at different moments during the entire training process, while LeRaC is applied only during the first few training epochs. Aside from these technical differences, our experiments already include a comparison of the two strategies (LeRaC vs. Linear Warmup with Cosine Annealing), as also pointed out in the response to reviewer YyP2. Indeed, the CvT results presented in Table 2 show that introducing LeRaC brings consistent improvements. We thus conclude that our strategy is a viable and distinct alternative to learning rate warmup. The relationship to Adam [R5] is detailed in the reply to reviewer Z9wJ."
>
> The reviewer was happy with our clarification, stating that "The author addresses some of my concerns like the differences between this work and the given related works."
>
> We provided the following answer to reviewer YyP2: "Interaction of LeRaC with adaptive optimizers. Re: Our initial learning rates and scheduler are used independently of the optimizers. As shown in Table 1, we used different optimizers, opting in each case for the best optimizer for each baseline architecture (without LeRaC). We kept the same optimizers when introducing LeRaC. Our learning rate scheduler updates the learning rates at the beginning of every iteration. We did not observe any stability / interaction issues. Learning rate schedulers may disturb the proposed model. Re: Whenever a learning rate scheduler was used for training a model, we simply replaced the scheduler with LeRaC. For example, all the baseline CvT results are based on Linear Warmup with Cosine Annealing (LWCA), this being the recommended scheduler for CvT. When we introduced LeRaC, we simply deactivated LWCA. In general, we recommend deactivating other schedulers while using LeRaC for simplicity in avoiding stability issues. We will mention this limitation in the final version of the paper."
>
> Reviewer YyP2 did not further acknowledge reading our rebuttal.